# Improving Adversarial Robustness via Frequency Regularization

## Abstract

Deep neural networks (DNNs) are incredibly vulnerable to crafted, human-imperceptible adversarial perturbations. While adversarial training (AT) has proven to be an effective defense approach, the properties of AT for robustness improvement remain an open issue. In this paper, we investigate AT from a spectral perspective, providing new insights into the design of effective defenses. Our analyses show that AT induces the deep model to focus more on the low-frequency region, which retains the shape-biased representations, to gain robustness. Further, we find that the spectrum of a white-box attack is primarily distributed in regions the model focuses on, and the perturbation attacks the spectral bands where the model is vulnerable. To train a model tolerant to frequency-varying perturbation, we propose a frequency regularization (FR) such that the spectral output inferred by an attacked input stays as close as possible to its natural input counterpart. Experiments demonstrate that FR and its weight averaging (WA) extension could significantly improve the robust accuracy by $1.14\% \sim 4.57\%$ relative to the AT, across multiple datasets (SVHN, CIFAR-10, CIFAR-100, and Tiny ImageNet), and various attacks (PGD, C&W, and Autoattack), without any extra data.

## 1 Introduction

DNNs have exhibited strong capabilities in various application such as computer vision He et al. (2016), natural language processing Devlin et al. (2018), recommendation systems Covington et al. (2016), etc. However, researches in adversarial learning show that even well-trained DNNs are highly susceptible to adversarial perturbations Goodfellow et al. (2014); Szegedy et al. (2013). These perturbations are nearly indistinguishable to human eyes but can mislead neural networks to completely erroneous outputs, thus endangering safety-critical application. Among various defense methods for improving robustness Das et al. (2018); Mao et al. (2019); Zheng et al. (2020), adversarial training (AT) Madry et al. (2017), which feeds adversarial inputs into a DNN to solve a min-max optimization problem, proves to be an effective means without obfuscated gradients problems Athalye et al. (2018). Some recent results inspired by AT are also in place to further boost the robust accuracy: Zhang et al. (2019) identify a trade-off between standard and robust accuracies that serves as a guiding principle for designing the defenses. Wu et al. (2020) claim that the weight loss landscape is closely related to the robust generalization gap, and propose an effective adversarial weight perturbation method to overcome the robust overfitting problem Rice et al. (2020). Jia et al. (2022) introduce a learnable attack strategy to automatically produce the proper hyperparameters for generating the perturbations during training to improve the robustness.

On the other hand, frequency analysis provides a new lens on the generalization behavior of DNNs. Wang et al. (2020a) claim that convolutional neural networks (CNNs) could capture human-imperceptible high-frequency components of images for predictions. It is found that robust models have smooth convolutional kernels in the first layer, thereby paying more attention to low-frequency information. Yin et al. (2019) establish a connection between the frequency of common corruptions and model performance, especially for high-frequency corruptions. It views AT as a data augmentation method to bias the model toward low-frequency information, which improves the robustness to high-frequency corruptions at the cost of reduced robustness to low-frequency corruptions. Zhang & Zhu (2019) find that AT-CNNs are better at capturing long-range correlations such as shapes, and less biased towards textures than normally trained CNNs in popular object recognition datasets. Our

findings are similar, but we learn them from a spectral perspective. Wang et al. (2020b) state that perturbations mainly focus on the high-frequency information in natural images, and low-frequency information is more robust than the high-frequency part. It is claimed that developing a stronger association between low-frequency information with true labels makes the model robust. However, our study shows that building this connection alone cannot render the model adversarial robustness. The closest work to ours is Maiya et al. (2021), which discovers that the adversarial perturbation is data-dependent and analyses many intriguing properties of AT with frequency constraints. Our research goes one step further to show that the perturbation is also model-dependent, and explains why it behaves differently across the datasets and models. Besides, we propose a frequency regularization (FR) to improve robust accuracy.

These breakthroughs motivate us to zoom in on deeper AT analysis from a spectral viewpoint. Specifically, we obtain models with different frequency biases and study the distribution of their corresponding white-box attack perturbations across different datasets. We then propose a simple yet effective FR to improve the adversarial robustness and perform validation on multiple datasets. Our main contributions are:

- We find that AT facilitates the model to focus on robust low-frequency information, which contains the shape-biased representation to improve the robustness. In contrast, simply focusing on low-frequency information does not lead to adversarial robustness.

- We reveal for the first time that the white-box attack is primarily distributed in the frequencies where the model focuses on, and can adapt its aggressive frequency distribution to the model's sensitivity to frequency corruptions. This explains why white-box attacks are hard to defend.

- We propose a FR that enforces alignment of the outputs of natural and adversarial examples in the frequency domain, thus effectively improving the adversarial robustness.

## 2 PRELIMINARIES

Typically, AT updates the model weights to solve the min-max saddle point optimization problem:

$$\min_{\theta} \frac{1}{n} \sum_{i=1}^{n} \max_{\|\delta\|_p \leq \epsilon} \mathcal{L}\left(f_\theta\left(\mathbf{x}_i + \delta\right), y_i\right), \tag{1}$$

where $n$ is the number of training examples, $\mathbf{x}_i + \delta$ is the adversarial input within the $\epsilon$-ball (bounded by an $L_p$-norm) centered at the natural input $\mathbf{x}_i$, $\delta$ is the perturbation, $y_i$ is the true label, $f_\theta$ is the DNN with weight $\theta$, $\mathcal{L}(\cdot)$ is the classification loss, e.g., cross-entropy (CE).

We refer to the adversarially trained model as the *robust model* and the naturally trained model as the *natural model*. The accuracy achieved on natural and adversarial inputs is denoted as *standard accuracy* and *robust accuracy*, respectively. We define the high-pass filtering (HPF) with bandwidth $k$ as the operation that after a Fast Fourier Transform (FFT), only the $k \times k$ patch in the center (viz. high frequencies) is preserved, and all external values are zeroed, and then applies inverse FFT. Low-pass filtering (LPF) is defined similarly except that the low-frequency part is shifted to the center after FFT, to be preserved by the center $k \times k$ patch as in Yin et al. (2019).

## 3 EMPIRICAL OBSERVATION AND ANALYSES

### 3.1 FREQUENCY ATTENTION & LOW-FREQUENCY INFORMATION

**Attention to the Frequency Domain.**    Since the labels are inherently tied with the low-frequency information Wang et al. (2020a), to maintain high standard accuracy and explore the connection between the low-frequency information and adversarial robustness, we train models (denoted as L-models) with the natural inputs after the LPF with a bandwidth of 16 for multiple datasets (32 for Tiny ImageNet), cf. Table 1. Then, we feed natural inputs processed by LPF with different bandwidths into models to evaluate the accuracy, which reflects how much attention the models pay to low-frequency information. Results are shown in Table 1.

For natural models, the standard accuracy gradually increases as the bandwidth increases, indicating that the models utilize both low- and high-frequency information, consistent with the findings

Table 1: Top-1 accuracy(%) of natural, L- and robust ResNet18 models. The bandwidth row denotes the LPF bandwidth ($k$) applied to the inputs. The higher the value, the more information is retained (i.e., 32 or 64 means no filtering). The last column shows the robust accuracy against PGD-20 attack. Bold numbers indicate the best.

| Dataset | Bandwidth | 4 | 8 | 12 | 16 | 20 | 24 | 28 | 32 | PGD-20 |
|---|---|---|---|---|---|---|---|---|---|---|
| SVHN | Natural | 49.07 | 83.87 | 94.82 | 95.41 | 95.50 | 95.53 | **95.60** | 95.58 | 0.38 |
| | L-model | 49.29 | 90.11 | 94.99 | 95.64 | 95.69 | **95.71** | 95.70 | 95.69 | 0.41 |
| | Robust | 40.47 | 81.54 | 88.43 | 89.30 | 89.36 | 89.38 | 89.38 | **89.39** | **53.74** |
| CIFAR-10 | Natural | 12.60 | 17.93 | 26.67 | 46.71 | 80.58 | 90.31 | 92.94 | **94.43** | 0.0 |
| | L-model | 18.41 | 45.29 | 84.87 | 91.45 | 92.02 | **92.06** | 92.03 | 92.03 | 0.05 |
| | Robust | 29.74 | 57.82 | 73.25 | 78.63 | 80.71 | 81.41 | 81.79 | **81.98** | **51.66** |
| CIFAR-100 | Natural | 3.20 | 7.52 | 19.91 | 45.46 | 62.34 | 69.24 | 71.33 | **74.95** | 0.0 |
| | L-model | 4.74 | 22.29 | 60.38 | 68.92 | **69.31** | 68.56 | 68.20 | 67.94 | 0.0 |
| | Robust | 13.99 | 33.94 | 45.30 | 50.27 | 52.56 | 53.42 | 54.1 | **54.18** | **27.7** |
| Dataset | Bandwidth | 8 | 16 | 24 | 32 | 40 | 48 | 56 | 64 | PGD-20 |
| Tiny ImageNet | Natural | 1.52 | 2.98 | 9.47 | 18.42 | 33.61 | 48.50 | 56.74 | **60.48** | 0.0 |
| | L-model | 2.78 | 4.97 | 35.48 | **56.80** | 56.28 | 55.77 | 55.88 | 55.76 | 0.0 |
| | Robust | 6.36 | 14.99 | 24.26 | 32.18 | 38.12 | 42.57 | 45.35 | **46.64** | **23.33** |

of Wang et al. (2020a). SVHN is an exception, as the information in this dataset is mainly concentrated in low-frequency region Bernhard et al. (2021). So the models trained on SVHN all primarily rely on the low-frequency information for predictions. For L-models, when the bandwidth increases to beyond 16 (32 for Tiny ImageNet), the standard accuracy no longer improves much, which is in line with the expectation that a model trained with low-frequency information focuses mainly on the low-frequency region for predictions. As for robust models, even though a large amount of high-frequency information is removed, there is only a negligible reduction in standard accuracy in the SVHN and CIFAR datasets. For Tiny ImageNet, the high-frequency content can further improve the standard accuracy a little, but the accuracy improvement relies mainly on low-frequency part.

Comparing the natural, L- and robust models across different datasets leads to the conclusion that AT enforces the AT-trained model to focus primarily on low-frequency information. Taking the CIFAR-10 as an example, the standard accuracy (81.98%) of the robust model is similar to that of the natural model at a LPF bandwidth of 20 (80.58%). Such observation indicates that the low standard accuracy of natural dataset in the robust model is due to the under-utilization of high-frequency components.

**Robust Features in the Low-frequency Region.** Although both the L-model and robust model rely on low-frequency information for predictions, the L-model has no resistance to PGD-20 (last column in Table 1), suggesting that learning the low-frequency information alone *does not* contribute to robust accuracy. Besides, when the input retains limited low-frequency information, the robust model is more accurate than the natural model, and even more accurate than the L-model in particularly small bandwidth (e.g., 4, 8) except for SVHN. This implies the robust model can extract more useful information from the particularly low-frequency region. To explore what robust information the model is concerned with, we visualize the images after LPF with a small bandwidth of 8, the natural images, and the perturbed images of the CIFAR-10 dataset in Fig. 1. Other datasets show similar performance, as depicted in Appendix A.1.

At $k = 8$, the robust model achieves a much higher standard accuracy (57.82%) than the natural model (17.93%) with very limited low-frequency information. Compared to natural images, the filtered images retain the outer contours, but the detailed textures are heavily blurred. And for the perturbed images, the texture of the foreground and background is disturbed, while the shapes are almost unaffected. Indeed, Geirhos et al. (2018) show that naturally trained CNNs are strongly biased toward recognizing textures rather than shapes, which accounts for the low standard accuracy for the filtered images and their vulnerability. The robust model can maintain a certain level of standard and robust accuracies, when the texture is heavily smoothed whereas the shape profile is partially preserved. This indicates that AT enables the model to learn a more shape-biased representation, which is more human-like and consistent with the finding of Zhang & Zhu (2019): AT-CNNs focus more on shape information. Learning shape-biased features improves robustness, and low-frequency information preserves objects' shapes while blurring textures. Consequently, AT induces models to

primarily focus on low-frequency information to learn a more shape-biased representation, which gains the robustness.

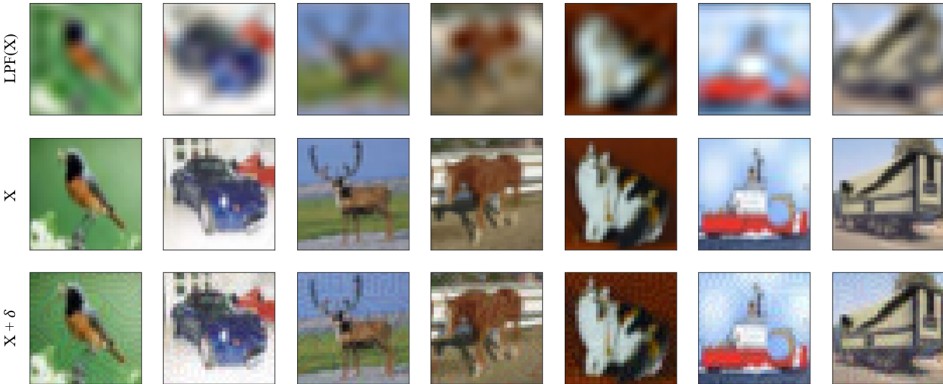

Figure 1: Visualization of the CIFAR-10 images after LPF with a bandwidth of 8 (top), natural images X (middle), and the perturbed images X+$\delta$ (bottom).

The results shown above indicate two points: **1)** AT induces the model to focus more on low-frequency region to learn a more shape-biased representation for improved robustness. **2)** Simply paying more attention to the low-frequency region without recognizing shapes does not yield adversarial robustness, which explains why some data augmentation schemes applied to the low-frequency region do not lead to a higher robustness Yin et al. (2019).

### 3.2 FREQUENCY DISTRIBUTION OF PERTURBATIONS

**Frequency Distribution of Perturbations.** Frequency analysis provides a new perspective on the understanding of network behaviors. A deep understanding of the perturbation's frequency distribution can provide new insights towards the design of effective defensive methods. Some defenses have been proposed motivated specifically by the hypothesis that adversarial perturbations lie primarily in the high-frequency region. However, other researches Bernhard et al. (2021); Maiya et al. (2021) have refuted this hypothesis and claimed the frequency distribution is related to the dataset.

To this end, we use the PGD attack with the maximum perturbation $\epsilon = 8/255$ as a representative of the white-box attack and analyse the frequency distribution of the perturbations. Since the generation of adversarial examples is determined by two main components, the target model and the input images, we propose a hypothesis that *the frequency distribution of the perturbations is related not only to the dataset but also to the model's frequency bias*. To verify this hypothesis, we take 2000 test examples to compute the average Fourier spectra of the PGD-20 attack perturbation for the natural, L- and robust models. The results are shown in Fig. 2.

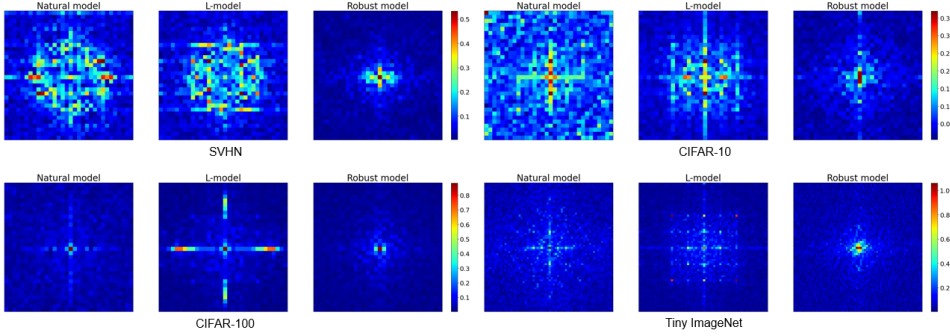

Figure 2: Visualization of perturbations in the frequency domain, with low frequency in the center.

Across the datasets, for the natural and L-models, the frequency distribution of the perturbations differ significantly. Robust models bias the perturbations towards the particular low regions, which is consistent with the observation that robust models extract more information from particularly low-frequency parts. The frequency distribution of the L-model is more concentrated in the low-frequency region compared to the natural one. There are clear square contours in the CIFAR-10 and Tiny ImageNet datasets, indicating that the attacks are mainly concentrated within the central square. High-amplitude values are largely concentrated in the centre for robust models. These observations prove that the perturbation's frequency distribution is related to both the dataset and the model's frequency bias. Frequency distributions of adversarial perturbations of randomly selected images across the datasets are shown in Appendix A.4.

**Aggressiveness of Perturbation.** To further explore the perturbation's aggressiveness of different frequency bands, we apply LPF and HPF to the perturbations and then add it to the natural examples to check the robust accuracy performance. The greater the drop in robust accuracy, the more aggressive the perturbation is in that spectral band. Besides, we add standard accuracy curves for natural examples filtered by LPF with different bandwidth to reflect the robust model's attention to the low-frequency information. The results are shown in Fig. 3.

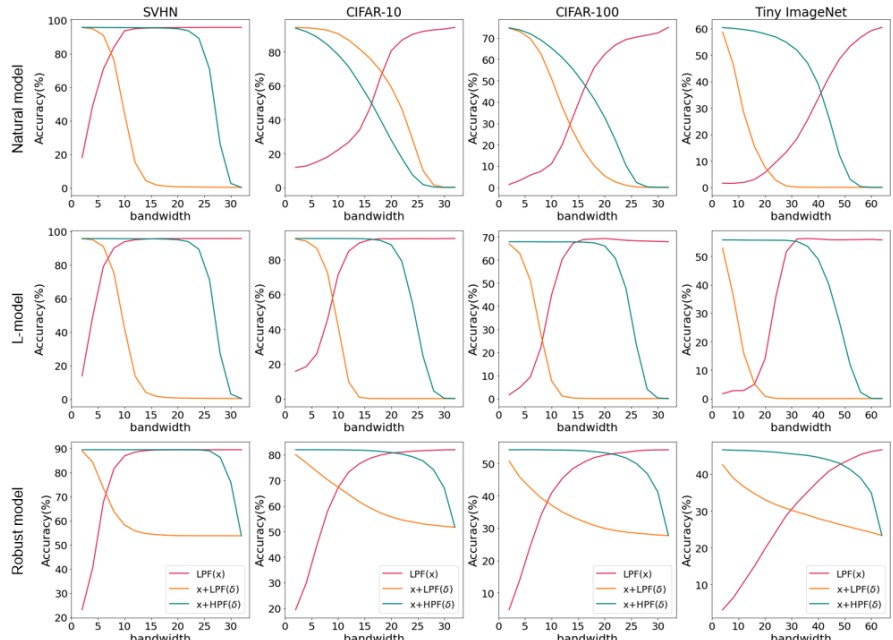

Figure 3: Standard accuracy and robust accuracy against PGD-20 attacks for natural, L- and robust ResNet18 models across different datasets. As the bandwidth increases, more input or perturbation information is retained.

SVHN is a unique dataset wherein the information is primarily concentrated in the low-frequency region. The models obtained on it by either training method rely mainly on low-frequency information for predictions. Perturbations also rely on low-frequency information to maintain their aggressiveness, while high-frequency perturbations barely degrade robust accuracy.

For the natural models of CIFAR and Tiny ImageNet datasets, whether the perturbations are processed by LPF or HPF, the robust accuracy decreases as the bandwidth increases until it reaches almost zero. It indicates that perturbations maintain their aggressiveness in both low- and high-frequency parts, which corresponds to the fact that natural models use both low- and high-frequency information for predictions. In CIFAR-10, the perturbation after HPF (green curve) leads to more accuracy degradation compared to LPF (orange curve) at the same bandwidth, which means the high-frequency perturbation is more aggressive. The opposite is true in the CIFAR-100 and Tiny ImageNet. In particular, in Tiny ImageNet, the aggressiveness of the perturbation is mainly concentrated at low frequencies. For natural models utilizing low- and high-frequency information, why is

the distribution of perturbation aggressiveness so different across different datasets? *We propose a sensitivity hypothesis that white-box attacks can detect spectral bands where the model is sensitive and formulate the attack correspondingly.*

To prove this assumption, we investigate the sensitivity of models to frequency corruptions via the Fourier heat maps Yin et al. (2019) shown in Fig. 4. The definition is described in Appendix A.2. A high error rate means that the model is vulnerable to the attacks with the corresponding frequency. The first row of Fig. 4 shows the Fourier heat maps of natural models across different datasets. The CIFAR-10 and CIFAR-100 are sensitive to the low- and high-frequency perturbations, which is consistent with the phenomenon that the perturbations after LPF or HPF can significantly degrade the robust accuracy. Tiny ImageNet is much more vulnerable to the low-frequency perturbations, which corresponds to the fact that perturbation aggressiveness is mainly concentrated at low frequencies. These phenomena validate our assumption: white-box attacks can detect a model-sensitive band of the spectrum and attack it.

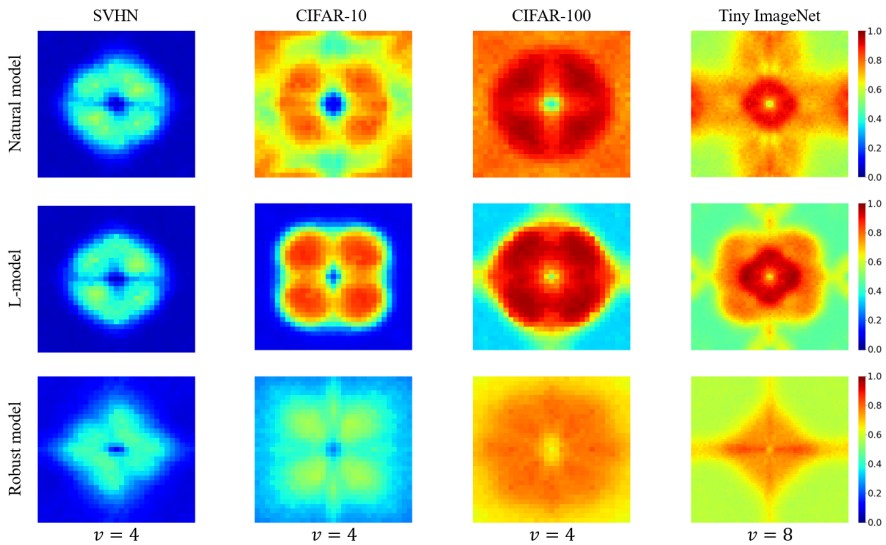

Figure 4: Error rate of models on the images perturbed with the spectral perturbations. We evaluate three models on Four datasets. $v$ is the norm of the perturbation. The high error rate represents the high sensitivity to the spectral noise.

For the L-models shown in Fig. 3, which extract information from the low-frequency region, the perturbation relies heavily on low-frequency information to ensure the success of attacks. As for the robust models, there is somewhat symmetry between the red and orange curves. In the CIFAR dataset, the model relies mainly on low-frequency information for predictions, and the perturbation similarly relies on low-frequency parts to degrade the robust accuracy. For the Tiny ImageNet dataset, the model can improve standard accuracy by a small margin with the help of high-frequency information. There is a small decrease in robust accuracy in both the high bandwidth region for LPF and the low bandwidth region for HPF of the perturbation, indicating that the perturbation is somewhat aggressive at high frequencies. The aggressive frequency distribution of the perturbations basically corresponds to the model's attention to the frequency domain. The L-models and robust models are sensitive to the low-frequency perturbations shown in the 2nd and 3rd rows of Fig. 4, which again concurs with our sensitivity hypothesis.

Comparing the Fourier heat maps of the natural and robust models on multiple datasets, it can be concluded that the robust model is less sensitive to spectral perturbations. This suggests methods that reduce the model's sensitivity to frequency corruptions should help improve adversarial robustness. Based on the above experiments, we claim that the white-box attacks are primarily distributed in the frequency domain where the model attends to, and can adjust their aggressive frequency distribution according to the model's sensitivity to frequency corruptions. In short, the white-box attacks can strike the frequency regions where the model's defenses are weak. Indeed, this is a first-ever spectral perspective to explain why white-box attacks are so hard to defend.

## 4 Frequency Regularization

Although AT improves robustness, there is still a large gap between standard and robust accuracies. Kannan et al. (2018) propose an adversarial logit pairing that forces the logits of a paired natural and adversarial examples to be similar. Zhang et al. (2019) utilize the classification-calibrated loss to minimize the difference between the prediction of natural and adversarial inputs. Bernhard et al. (2021) apply LPF and HPF to the inputs, and then minimize the output difference between natural and filtered inputs. Tack et al. (2022) improve the robustness by forcing the predictive distributions after attacking from two different augmentations of the same input to be similar.

For the first time in literature, we have demonstrated in Section 3.2 that the frequency distribution of a perturbation is related to both the dataset and model, which is not a simple low- or high-frequency phenomenon, and the white-box attack can adapt its aggressive frequency distribution to the target model. Intuitively, a natural idea is to drive the model to limit or tolerate this spectral difference between the outputs subject to a natural input and its adversarial counterpart, and to achieve similar frequency-domain outputs for both types of inputs. By updating the weights through the back-propagation mechanism, this constraint makes the model extract similar spectral features from the adversarial inputs as the natural inputs. Then the robust accuracy will gradually approach the standard accuracy and thus be improved. To achieve this goal, we devise a simple yet effective frequency regularization (FR) to align the difference of the outputs between the natural and adversarial inputs in the frequency domain, as shown in Fig. 5. The optimization goal of the proposed AT with FR is:

$$\mathcal{L}_{AT} = \mathcal{L}_{CE} + \lambda \cdot \frac{1}{n} \sum_{i=1}^{n} Dis(\mathcal{F}(f_1(x_i)), \mathcal{F}(f_2(x_i + \delta))), \qquad (2)$$

where $\lambda$ (defaulted at 0.1) denotes the FR coefficient, $f_1$, $f_2$ are the DNNs (same model for the basic FR) and $f_2$ is used for prediction, $Dis$ denotes the distance function ($\mathcal{L}_1$ norm is used), $\mathcal{L}_{CE}$ is the Cross-Entropy loss and $\mathcal{F}$ denotes FFT. The distance function is applied to the real and imaginary parts of the complex numbers after FFT, respectively, and the results are summed. FR consists of two branches, one dealing with natural inputs and the other with adversarial inputs. Because the standard accuracy is higher than the robust accuracy, it may reduce the standard accuracy while increasing the robustness. To control the degradation of standard accuracy while maintaining robustness, we need to find the proper model to handle natural inputs.

Weight averaging (WA) Izmailov et al. (2018) depicted in Appendix A.3, which averages the weight values over epochs along the training trajectory, proves to be an effective means to improve the generalization of models. In AT, it could be combined with other methods Gowal et al. (2020); Chen et al. (2020) to mitigate the robust overfitting problem Rice et al. (2020). These works use WA to generate the final model for evaluation. During AT, the WA model maintains a similar standard and robust accuracies to the current training model, and its weights are fixed. If we utilize the WA model to deal with natural inputs, the branch of FR/WA processing natural inputs (cf. WA branch in Fig.5) will not update the weights to force the standard accuracy approach the robust accuracy. Therefore, instead of using the WA model for the final evaluation, we utilize the WA model to deal with natural inputs. In simple terms, we replace the $f_1$ model in Eqn. 2 with the WA model generated during AT.

## 5 Experiments

**Datasets.** Without loss of generality, we select four common image datasets: SVHN Netzer et al. (2011), CIFAR-10, CIFAR-100 Krizhevsky et al. (2009) and Tiny ImageNet. We apply 4-pixel padding with $32 \times 32$ random crop (not for SVHN and Tiny ImageNet) and random horizontal flip (not for SVHN) for data enhancement. All natural images are normalized to $[0, 1]$. SVHN, CIFAR-10, and CIFAR-100 image resolution is $32 \times 32 \times 3$, corresponding to the length, width, and channel, respectively. Tiny ImageNet image resolution is $64 \times 64 \times 3$.

**Experimental Settings.** We take ResNet18 as a default model and adopt a SGD optimizer with a momentum of 0.9 and a global weight decay of $5 \times 10^{-4}$. The model is trained with PGD-10 for 100 (30)[1] epochs with a batch size of 128 on one 3090 GPU. The initial learning rate is 0.1 (0.01), which

---

[1]The numbers in brackets are the hyperparameters for SVHN.

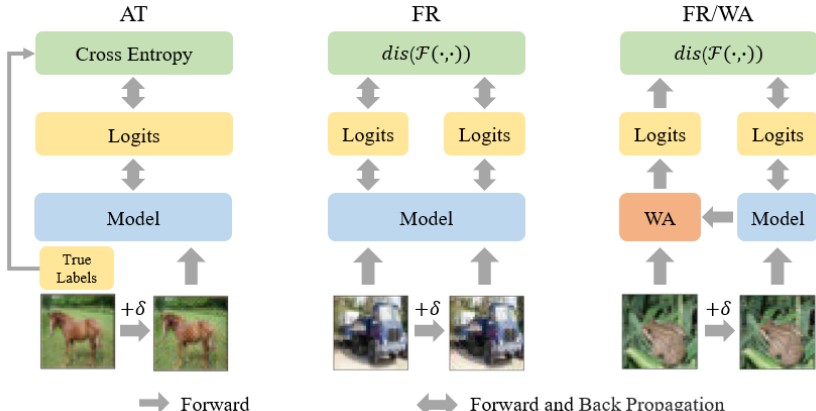

Figure 5: An overview of the standard AT, the proposed FR, and its WA extension. $\delta$ denotes the perturbation, $\mathcal{F}$ denotes the FFT, *dis* denotes the distance function.

decays to one-tenth at 75th (15th) and 90th (25th) epochs, respectively. The robust accuracy of the PGD-20 attack equipped with a random-start is taken as the main basis for robustness analysis. The attack step size is $\alpha$ = 2/255 and maximum $l_\infty$ norm-bounded perturbation $\epsilon$ = 8/255. FR and WA are used since the first epoch where the learning rate drops, and continues until the end with a cycle length 1. We also show our method is suitable for large-scale models in Appendix A.5.

**Evaluated Attacks.** The model with the highest robust accuracy against PGD-20 is selected for further evaluation. To avoid a false sense of security caused by the obfuscated gradients, we evaluate the robust accuracy against several popular white-box attack methods, including PGD Madry et al. (2017), C&W Carlini & Wagner (2017), and AutoAttack Croce & Hein (2020) (denoted as AA, consists of APGD-CE, APGD-DLR, FAB, and Square). Following the default setting of AT, the attack step size is 2/255, and the maximum $l_\infty$ norm-bounded perturbation is 8/255.

### 5.1 ABLATION STUDIES

**Distance Function in FR.** For the distance function defines in Eqn. 2, there are three frequently used methods, including $\mathcal{L}_1$ norm, $\mathcal{L}_2$ norm, and cosine similarity. We measure their performance on CIFAR-10. For fair comparison, we use the same checkpoint from the 74th epoch and then apply the different distance functions, respectively. As shown in Figs. 6(a) and 6(b), in terms of improving robustness, $\mathcal{L}_1$ norm is more effective at the expense of a slightly reduced standard accuracy.

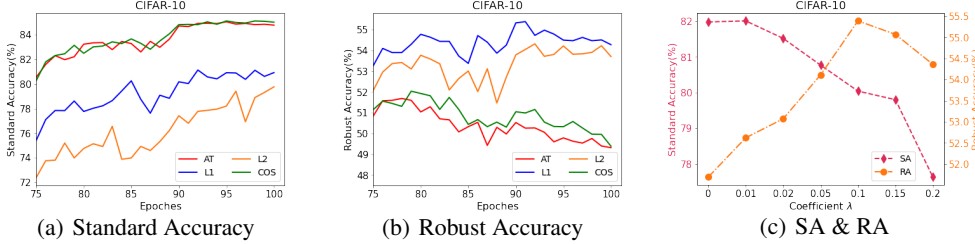

     (a) Standard Accuracy         (b) Robust Accuracy          (c) SA & RA

Figure 6: Ablation of distance functions and coefficient $\lambda$ on CIFAR-10. SA and RA denote the standard and robust accuracy. (a) and (b) show the SA and RA of different distance functions, AT represents the standard AT without FR. (c) shows the impact of $\lambda$ on SA and RA.

**Sensitivity to Regularization Coefficient.** We further investigate the impact of the parameter $\lambda$ in FR defined in Eqn. 2 which modifies the strength of the regularization. The results for different $\lambda \in [0, 0.2]$ are shown in Fig. 6(c). 0 represents the standard AT. Results show a trade-off between the standard and robust accuracies when the $\lambda$ is within a proper limit $[0, 0.1]$. When $\lambda$ is outside this range, FR dominates the loss function, resulting in an overall decrease in accuracy.

Table 2: Top-1 robust accuracy(%) against diverse attacks with maximum $l_\infty$ norm-bounded perturbation $\epsilon = 8/255$ of ResNet18 models. Bold numbers indicate the best on different datasets.

| Dataset | Method | Clean | PGD-20 | PGD-50 | C&W | AA |
|---|---|---|---|---|---|---|
| SVHN | PGD-AT | **90.88** | 53.28 | 52.26 | 50.62 | 47.57 |
| | AT+FR | 90.43 | 56.87 | 56.29 | **51.89** | **49.46** |
| | AT+FR/WA | 90.49 | **56.95** | **56.35** | 51.76 | 49.36 |
| CIFAR-10 | PGD-AT | **81.98** | 51.69 | 51.46 | 50.44 | 48.19 |
| | AT+FR | 80.04 | **55.39** | **55.13** | 51.61 | 50.02 |
| | AT+FR/WA | 81.74 | 55.12 | 54.88 | **52.21** | **50.16** |
| CIFAR-100 | PGD-AT | **54.18** | 27.81 | 27.49 | 25.82 | 23.77 |
| | AT+FR | 49.23 | 31.27 | 31.20 | 27.60 | **26.09** |
| | AT+FR/WA | 53.66 | **31.49** | **31.36** | **28.24** | 26.06 |
| Tiny ImageNet | PGD-AT | **46.64** | 23.33 | 23.18 | 20.44 | 18.34 |
| | AT+FR | 42.92 | 25.32 | 25.25 | 21.35 | **19.71** |
| | AT+FR/WA | 46.55 | **25.64** | **25.47** | **21.90** | 19.64 |

Table 3: Top-1 robust accuracy(%) of the WideResNet-34-10 model on the CIFAR-10. Bold numbers indicate the best.

| Method | Clean | PGD-20 | PGD-50 | C&W | AA |
|---|---|---|---|---|---|
| PGD-AT Rice et al. (2020) | 84.62 | 55.01 | 54.88 | 53.32 | 51.42 |
| TRADES Zhang et al. (2019) | 84.92 | 56.33 | 56.13 | 54.20 | 53.08 |
| MART Wang et al. (2019) | 84.17 | 58.56 | 58.06 | 54.58 | 51.10 |
| AWP Wu et al. (2020) | **85.57** | 58.14 | 57.92 | 55.96 | 54.04 |
| AT + FR (ours) | 82.66 | **59.38** | **59.15** | 55.72 | 54.33 |
| AT + FR/WA (ours) | 84.57 | 59.27 | 58.90 | **56.92** | **54.80** |

## 5.2 EXPERIMENTAL RESULTS AND ANALYSES

**Superior Performance across Datasets.** As shown in Table 2, we incorporate the FR and FR/WA into AT to improve the robustness against various attacks on multiple datasets. In particular, for AT, the FR/WA version relatively improves 3.27% and 1.84% of robust accuracy on average against the PGD-20 and AA, respectively, with a much smaller degradation (0.31%) in standard accuracy. These results indicate that our methods are versatile across various datasets. Experiments in Appendix A.6 indicate the FR and FR/WA can be plugged into other defenses to further improve the robustness.

**Benchmark with Other Defenses.** Table 3 further compares the impact of FR with famous defenses (details of the defenses are reviewed in the Appendix A.7) on the CIFAR-10 dataset. Wide ResNet-34-10 Zagoruyko & Komodakis (2016) is the popular model for comparison. The results show that the FR and FR/WA substantially improve robust accuracy compared to the AT and outperform other defenses. Besides, FR/WA improves robustness while maintaining standard accuracy.

## 6 CONCLUSION

This work explores the appealing properties of adversarial perturbation and AT from a spectral lens. We find that AT renders the model more focused on shape-biased representation in the low-frequency region to gain robustness. Using systematic experiments, we show for the first time that the white-box attack can adapt its aggressive frequency distribution to the target model's sensitivity to frequency corruptions, making it hard to defend. To enhance tolerance to frequency-varying perturbations, we further devise a frequency regularization (FR) to align the outputs with respect to natural and adversarial inputs in the spectral domain. Experiments show that FR can substantially improve robust accuracy without extra data. It is believed these novel insights can advance our knowledge about the frequency behavior of AT and shed more light on robust network design.

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

# A APPENDIX

## A.1 VISUALIZATION OF THE FILTERED, NATURAL AND PERTURBED IMAGES FROM DIFFERENT DATASETS.

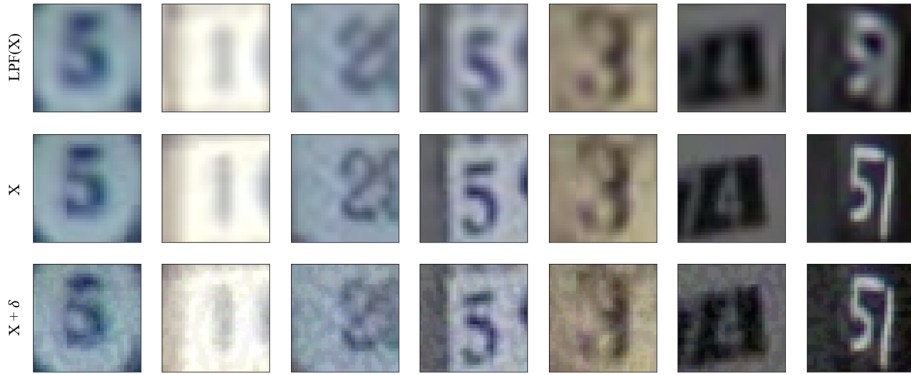

Figure 7: Visualization of the SVHN images after LPF with a bandwidth of 8 (top), natural images X (middle), and the perturbed images X+$\delta$ (bottom).

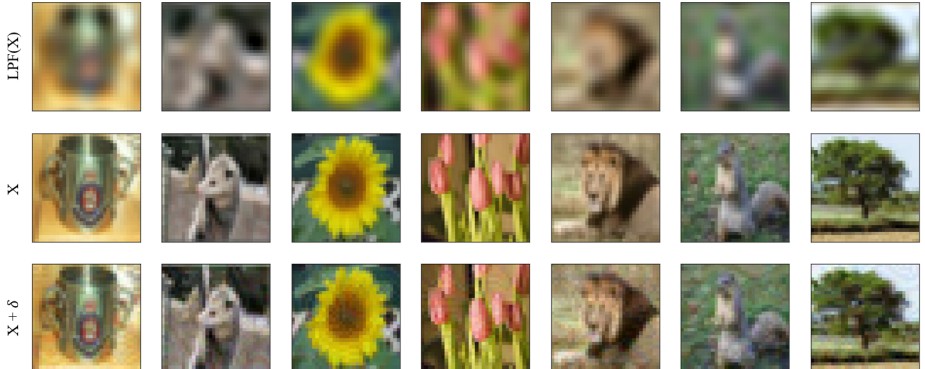

Figure 8: Visualization of the CIFAR-100 images after LPF with a bandwidth of 8 (top), natural images X (middle), and the perturbed images X+$\delta$ (bottom).

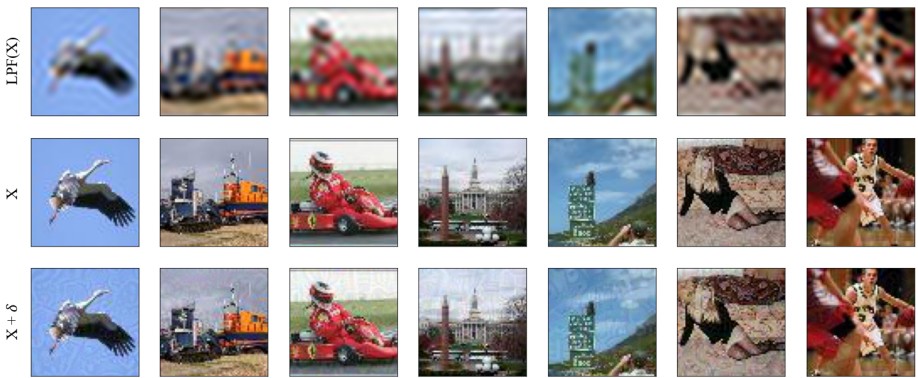

Figure 9: Visualization of the Tiny ImageNet images after LPF with a bandwidth of 8 (top), natural images X (middle), and the perturbed images X+$\delta$ (bottom).

## A.2 Fourier Heat Map

Fourier heat map provides a perturbation analysis method to investigate the sensitivity of models to the frequency corruptions. More precisely, let $U_{i,j} \in \mathbb{R}^{d_1 \times d_2}$ be a real-valued matrix such that $\|U_{i,j}\|_2 = 1$, and $FFT(U_{i,j})$ only has up to two non-zero elements located at $(i, j)$ and its symmetric coordinate with respect to the center. These matrices are denoted as 2D *Fourier basis* matrices. Given a model, we can generate the perturbed image $\widetilde{X} = X + rvU_{i,j}$ from the natural image $X$, where $r$ is chosen uniformly at random from $\{-1, 1\}$, and $v$ is the norm magnitude of the perturbation. For multi-channel images, we perturb every channel independently. We can then calculate the error rate of the model under *Fourier basis* noises and visualize how the error rate changes as a function of the spectral indices. The visualization result is called a Fourier heat map. In this paper, We move the low-frequency region to the center of the image. A high error rate means the model is vulnerable to attacks with the corresponding frequency.

## A.3 Weight Averaging

Following the definition in Izmailov et al. (2018), the equation of WA is:

$$\mathcal{W}_{wa}^n = \frac{\mathcal{W}_{wa}^{n-1} \times k + \mathcal{W}^n}{k + 1} \tag{3}$$

where $k$ denotes the number of past checkpoints to be averaged, $n$ denotes the index of the epoch during the training, $\mathcal{W}_{wa}^n$ denotes the weights of the WA model at $n$-th epoch, $\mathcal{W}^n$ denotes the current model's weights. In this paper, The WA model is viewed as a teacher dealing with the natural inputs, and the model that we evaluate the accuracy is viewed as a student. We hope the teacher (WA) helps the student extract useful information from the perturbed images, and we evaluate the robustness of the student model *not* on the teacher (WA) model.

## A.4 Adversarial Perturbations and Spectral Distribution

Figures 10–13 show the natural images, adversarial images, and the spectral distribution (low frequency in the center) of the perturbations across the datasets. $x$ denotes the natural images, $\delta_{nm}$, $\delta_{lm}$ and $\delta_{rm}$ denote the PGD-20 attack perturbations generated according to the natural, L- and robust models, respectively. $FFT$ denotes the Fast Fourier Transform. Jet color map is used to highlight perturbations for clear visualization. For the natural models, the perturbations are a jumble of noise points within the picture and have large magnitudes in the high-frequency region. The perturbations for the adversarial models are significantly more ordered and mainly concentrated in the low-frequency region. These visualizations prove that the adversarial perturbation is *not* a simple high-frequency phenomenon and is model- and dataset-dependent.

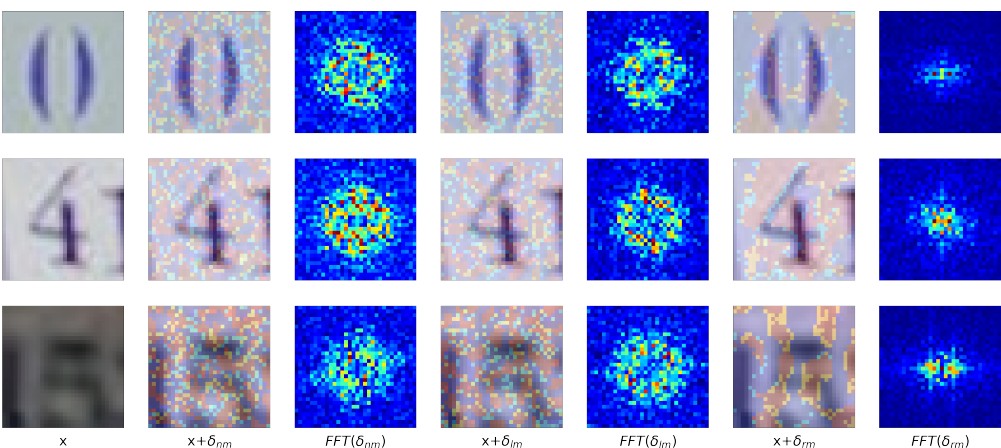

Figure 10: Visualization of the natural and perturbed images on SVHN.

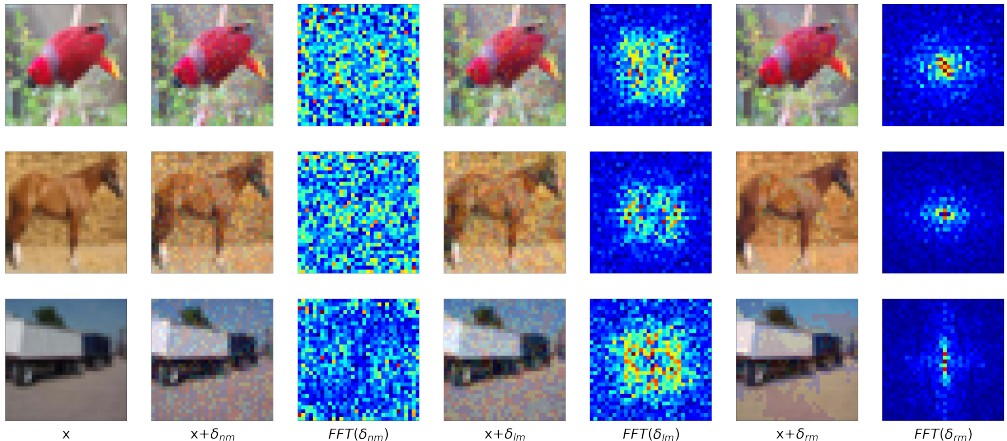

Figure 11: Visualization of the natural and perturbed images on CIFAR-10.

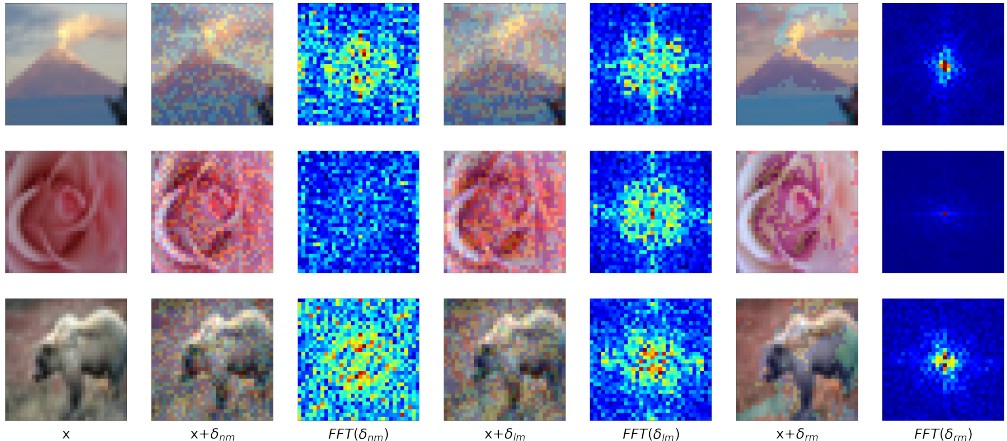

Figure 12: Visualization of the natural and perturbed images on CIFAR-100.

### A.5 LARGE ARCHITECTURE

To prove the effectiveness of the FR and FR/WA blocks on large models, we trained the ResNet152 and Wide ResNet-34-15 on the CIFAR-10 dataset. The results are shown in Table 4. For reference, the ResNet18 and Wide ResNet-34-10 have a parameter count of 11.174M and 46.160M, respectively. For large models, FR and FR/WA can still improve the model's robustness against multiple attacks relative to the standard AT, proving that the proposed methods are suitable for large models.

### A.6 MORE COMPARISONS WITH FR

**Experimental setup:** For a fair comparison, all experiments adopt the same data augmentation method: 4-pixel padding with $32 \times 32$ random crops (not for SVHN and Tiny ImageNet) and random horizontal flip (not for SVHN). All natural images are normalized to $[0, 1]$. The Frequency Regularization (FR) coefficient is set to 0.1 for SVHN and CIFAR datasets, and 0.05 for Tiny ImageNet. The training set was randomly divided into the training set and the validation set according to the ratio of 9:1. We select the model with the highest robustness against PGD-20 attacks on the validation set for further evaluation against other popular attacks.

**Effectiveness across the datasets and methods:** We provide the thorough performance comparison of AT+FR and other methods on ResNet18 in Table 5-8. Since the FR and FR/WA are plug-and-

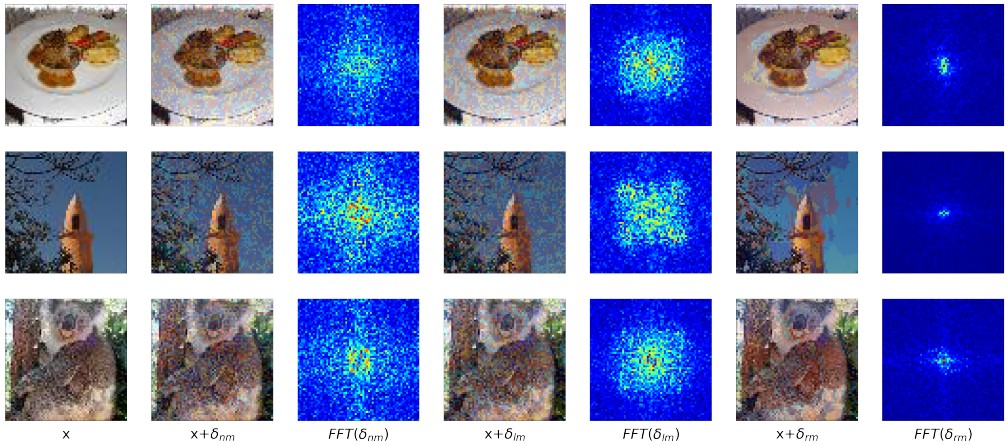

$x$    $x+\delta_{nm}$    $FFT(\delta_{nm})$    $x+\delta_{lm}$    $FFT(\delta_{lm})$    $x+\delta_{rm}$    $FFT(\delta_{rm})$

Figure 13: Visualization of the natural and perturbed images on Tiny ImageNet.

Table 4: Top-1 accuracy(%) of various models on the CIFAR-10. #number indicates the parameters. Bold numbers indicate the best.

| model | Method | Clean | PGD-20 | PGD-50 | C&W | AA |
|---|---|---|---|---|---|---|
| ResNet152 #58,156,618 | AT | 84.60 | 54.62 | 54.44 | 52.86 | 50.87 |
| | AT + FR | 81.11 | **57.76** | **57.70** | 54.74 | 53.32 |
| | AT + FR/WA | **84.62** | 57.55 | 57.50 | **55.63** | **54.10** |
| Wide ResNet-34-15 #103,819,674 | AT | **86.66** | 55.79 | 55.49 | 54.58 | 52.76 |
| | AT + FR | 84.35 | **58.33** | **58.03** | **55.75** | **54.12** |
| | AT + FR/WA | 86.18 | 57.97 | 57.78 | 55.25 | 53.50 |

play blocks, we also apply them to popular techniques (detailed experimental settings are depicted in Appendix A.7) to prove their effectiveness. Experimental results demonstrate that FR can be plugged into these popular methods to further improve robustness. Besides, FR/WA can maintain a similar standard accuracy as AT while improving robust accuracy. The improvement is non-trivial since some papers have claimed a trade-off between the standard and robust accuracy.

Table 5: Top-1 accuracy(%) of the ResNet18 model on the SVHN. Bold numbers indicate the best.

| Method | Clean | PGD-20 | PGD-50 | C&W | AA |
|---|---|---|---|---|---|
| AT | **90.88** | 53.28 | 52.26 | 50.62 | 47.57 |
| AT + FR | 90.43 | 56.87 | 56.29 | **51.89** | **49.46** |
| AT + FR/WA | 90.49 | **56.95** | **56.35** | 51.76 | 49.36 |
| TRADES | **89.04** | 55.71 | 55.48 | 51.48 | 50.03 |
| TRADES + FR | 87.23 | 56.96 | 56.73 | **52.64** | **51.26** |
| TRADES + FR/WA | 88.76 | **57.01** | **56.76** | 52.53 | 51.18 |
| MART | **88.67** | 56.78 | 56.40 | 50.03 | 47.75 |
| MART + FR | 86.42 | **57.87** | **57.52** | **50.65** | **48.74** |
| MART + FR/WA | 88.53 | 57.58 | 57.26 | 50.42 | 48.58 |
| AWP | **89.12** | 53.85 | 53.23 | 49.84 | 47.05 |
| AWP + FR | 86.63 | **55.64** | **55.33** | 50.83 | **49.53** |
| AWP + FR/WA | 88.73 | 55.32 | 55.11 | 50.68 | 49.37 |

Table 6: Top-1 accuracy(%) of the ResNet18 model on the CIFAR-10. Bold numbers indicate the best.

| Method | Clean | PGD-20 | PGD-50 | C&W | AA |
|---|---|---|---|---|---|
| AT | **81.98** | 51.69 | 51.46 | 50.44 | 48.19 |
| AT + FR | 80.04 | **55.39** | **55.13** | 51.61 | 50.02 |
| AT + FR/WA | 81.74 | 55.12 | 54.88 | **52.21** | **50.16** |
| TRADES | **81.83** | 53.41 | 53.23 | 50.92 | 49.84 |
| TRADES + FR | 80.11 | **54.45** | **54.24** | **51.51** | **50.52** |
| TRADES + FR/WA | 81.76 | 54.14 | 54.06 | 51.31 | 50.38 |
| MART | **81.01** | 54.58 | 54.47 | 50.01 | 48.10 |
| MART + FR | 79.03 | **55.17** | **54.90** | **50.98** | **49.22** |
| MART + FR/WA | 80.80 | 55.01 | 54.78 | 50.12 | 48.78 |
| AWP | **81.06** | 55.36 | 55.27 | 51.98 | 50.37 |
| AWP + FR | 79.03 | **57.07** | **57.01** | **52.16** | **50.80** |
| AWP + FR/WA | 80.87 | 56.81 | 56.82 | 52.14 | 50.61 |

Table 7: Top-1 accuracy(%) of the ResNet18 model on the CIFAR-100. Bold numbers indicate the best.

| Method | Clean | PGD-20 | PGD-50 | C&W | AA |
|---|---|---|---|---|---|
| AT | **54.18** | 27.81 | 27.49 | 25.82 | 23.56 |
| AT + FR | 49.23 | 31.27 | 31.20 | 27.60 | **26.09** |
| AT + FR/WA | 53.66 | **31.49** | **31.36** | **28.24** | 26.06 |
| TRADES | 56.24 | 28.48 | 28.40 | 24.71 | 23.77 |
| TRADES + FR | 55.57 | 30.08 | 29.95 | 25.93 | 25.05 |
| TRADES + FR/WA | **56.59** | **30.22** | **30.20** | **26.81** | **25.37** |
| MART | **51.23** | 29.66 | 29.55 | 25.88 | 24.27 |
| MART + FR | 49.33 | 31.03 | 30.87 | 26.78 | 25.07 |
| MART + FR/WA | 50.72 | **31.75** | **31.68** | **27.32** | **25.46** |
| AWP | 54.71 | 30.88 | 30.69 | 27.87 | 25.74 |
| AWP + FR | 48.92 | **31.90** | **31.73** | 28.02 | 26.10 |
| AWP + FR/WA | **55.78** | 31.75 | 31.60 | **28.84** | **26.64** |

## A.7 DETAILED SETTINGS FOR POPULAR DEFENSES

**TRADES Zhang et al. (2019):** It decomposes the robust error as the sum of the natural error and the boundary error and encourages the algorithm to push the decision boundary away from the data to improve the robust accuracy. The overall loss function is shown as follows:

$$\mathcal{L}_{AT} = \mathbf{CE}(f(x), y) + \lambda \cdot \mathbf{KL}(f(x) || f(x + \delta)) \quad (4)$$

**CE** denotes the Cross-Entropy loss, **KL** denotes the Kullback-Leibler divergence generated by PGD-10, $\delta$ denotes the adversarial perturbations, $f(x)$ denotes the probability predicted by the model, $y$ denotes the true label, $\lambda$ is the coefficient to balance the $CE$ and $KL$ loss. Following the default setting in TRADES, we adopt SGD with momentum 0.9, weight decay $2 \times 10^{-4}$, and batch size 128. The model is trained for 100 (30)[2]. epochs on one 3090 GPU. The initial learning rate is 0.1 (0.01), which decays to one-tenth at 75th (15th) and 90th (25th) epochs, respectively. The $\lambda$ in Eqn. 5 is set to 6.

**MART Wang et al. (2019):** Based on standard AT, it explicitly differentiates the misclassified and correctly classified examples during the training and adds a misclassification-aware regularization to the standard adversarial risk to achieve better robustness. The overall loss function is shown as follows:

$$\mathcal{L}_{AT} = \mathbf{BCE}(f(x + \delta), y) + \lambda \cdot \mathbf{KL}(f(x) || f(x + \delta)) \cdot (1 - f(x)) \quad (5)$$

---

[2]The numbers in brackets are the hyperparameters for SVHN.

Table 8: Top-1 accuracy(%) of the ResNet18 model on the Tiny ImageNet. Bold numbers indicate the best.

| Method | Clean | PGD-20 | PGD-50 | C&W | AA |
|---|---|---|---|---|---|
| AT | **46.64** | 23.33 | 23.18 | 20.44 | 18.34 |
| AT + FR | 42.92 | 25.32 | 25.25 | 21.35 | **19.71** |
| AT + FR/WA | 46.55 | **25.64** | **25.47** | **21.90** | 19.64 |
| TRADES | 48.33 | 22.77 | 22.71 | 18.89 | 17.95 |
| TRADES + FR | 46.61 | 23.79 | 23.72 | 19.97 | 18.99 |
| TRADES + FR/WA | **48.94** | **24.81** | **24.73** | **20.21** | **19.34** |
| MART | **45.39** | 24.17 | 24.09 | 20.07 | 18.49 |
| MART + FR | 43.35 | 25.35 | 25.31 | 20.63 | 19.31 |
| MART + FR/WA | 44.79 | **25.83** | **25.80** | **21.53** | **19.44** |
| AWP | **46.49** | 24.76 | 24.59 | 21.04 | 19.11 |
| AWP + FR | 44.93 | **24.98** | **24.90** | **21.54** | **19.52** |
| AWP + FR/WA | 46.32 | 24.86 | 24.77 | 21.32 | 19.38 |

**BCE** denotes the binary Cross-Entropy loss. Following the default setting in MART, we adopt SGD with momentum 0.9, weight decay $2 \times 10^{-4}$, and batch size 128. The model is trained for 100 (30) epochs on one 3090 GPU. The initial learning rate is 0.1 (0.01), which decays to one-tenth at 75th (15th) and 90th (25th) epochs, respectively. The $\lambda$ in Eqn. 5 is set to 6.

**AWP Wu et al. (2020):** It identifies the connection between the weight loss landscape and the robust generalization gap, proposes adversarial weight perturbation to directly make the weight loss landscape flat, and develops a double perturbation (adversarially perturbing both inputs and weights) mechanism in the AT framework. Following the default setting in AWP, we adopt SGD with momentum 0.9, weight decay $5 \times 10^{-4}$, and batch size 128. The model is trained for 200 epochs on two V100 GPUs. The initial learning rate is 0.1 (0.01), which decays to one-tenth at 100th and 150th epochs, respectively.

