# OpenReview forum: "Improving Adversarial Robustness via Frequency Regularization"
_ICLR.cc/2023/Conference — Submitted to ICLR 2023_

### Official Review · Reviewer_5QQs · 2022-10-23

**Confidence:** 4
**Correctness:** 2
**Technical Novelty And Significance:** 2
**Empirical Novelty And Significance:** 2
**Recommendation:** 5

**Clarity, Quality, Novelty And Reproducibility:**

Clarity,  the paper is well-organized and easy to follow.

Quality, the results illustrated in the paper is somehow preliminary (see detailed comments);

Novelty : the hypothesis that attack can adapt to the target model’s sensitivity in frequency domain seems novel and sound; although how to turn this finding into an effective defending approach remain to be explored;

Reproducibility: reproducible;

**Strength And Weaknesses:**


+ Overall the paper is well-organized and easy to follow.

+ Empirical investigation about the hypothesis is interesting and clearly illustrated in Fig. 2/3/4.

- The proposed FT seems not as effective as it should be.

    a. Improvements shown in Tab. 2/3 are only marginal, not substantial as the paper claimed.

    b. Experiments are rather limited (Tab. 2 only compare the proposed method with PGD-AT; Tab 3 only test WideResNet on CIFAR10), why not report thorough comparison results with more models / datasets / against all previous methods?

- The proposed FT in (2) is probably not a wise choice:

   a. naively minimizing L1 distance between frequency domain responses of natural and adversarial inputs  did not capture the notion of model vulnerability. Instead, distribution-based divergence is probably more appropirate.

   b. even if the proposed FT enforces the model extract similar spectral features, PGD-AT and other attacks may still exploit the remaining model vulnerability.  Authors need to investigate failure cases and show what type of perturbations attacks may still constructed;


**Summary Of The Paper:**

This paper hypothesizes that "the white-box attack can adapt its aggressive frequency distribution to the target model’s sensitivity
to frequency corruptions". Empirical investigations are illustrated in Fig. 2/3/4 to support the hypothesis.

Based on these observations, authors then introduces a frequency regularization (FT) to enforce "the model extract similar spectral features from the adversarial inputs as the natural inputs". Experimental results in Tab. 2./3 indicate the effectiveness of the proposed method.

**Summary Of The Review:**

+ Overall the paper is well-organized and easy to follow.

+ Empirical investigation about the hypothesis is interesting and clearly illustrated in Fig. 2/3/4.

- The proposed FT seems not as effective as it should be.

    a. Improvements shown in Tab. 2/3 are only marginal, not substantial as the paper claimed.

    b. Experiments are rather limited (Tab. 2 only compare the proposed method with PGD-AT; Tab 3 only test WideResNet on CIFAR10), why not report thorough comparison results with more models / datasets / against all previous methods?

- The proposed FT in (2) is probably not a wise choice:

   a. naively minimizing L1 distance between frequency domain responses of natural and adversarial inputs  did not capture the notion of model vulnerability. Instead, distribution-based divergence is probably more appropirate.

   b. even if the proposed FT enforces the model extract similar spectral features, PGD-AT and other attacks may still exploit the remaining model vulnerability.  Authors need to investigate failure cases and show what type of perturbations attacks may still constructed;

I'd like to see more convincing results to demonstrate how to turn the interesting finding into an effective defending approach in follow up work.

---

> ### Author Response · Authors · 2022-11-18
> **Response to Reviewer 5QQs**
>
> We thank R4 for finding our work interesting and for providing valuable feedback.
>
> 1. R4-1a (marginal improvement): We know there is still a large gap between the adversarial robustness and the standard accuracy, yet it is widely perceived that raising adversarial robustness is a highly difficult task. LAS-AT[1] utilizes reinforcement learning based on the standard AT. MART[2] revisited misclassified examples in AT. None of these techniques applied to the AT give as high a boost as the FR in this paper. To improve robustness significantly, sophisticated techniques[3] and more data[4] are required. Our proposed method is **low in complexity and does not require any additional data**. In particular, we proposed a **simple but effective** FR block. Taking the standard PGD-AT as the baseline, FR obtains higher adversarial robustness without bells and whistles compared with some SOTA methods, and no additional data is required. In addition, there is a trade-off between the standard and the robust accuracies. The FR/WA method can **maintain the standard accuracy as the PGD-AT while improving the robust accuracy**, which is something nontrivial.
>
> 2. R4-1b (Limited experiments): CIFAR-10 is the most commonly used dataset for AT studies, and some papers[1,2,3] have used Wide ResNet-34-10 as the baseline model for comparison. So we apply the same configuration in Table 3 to benchmark with other famous defenses. **Additional experimental results** are available in Section 3 of the supplementary. The conclusion is listed as follows:
>     - Due to time and resource constraints, we used ResNet18 to do comparative experiments. The results prove the proposed methods **perform better than these famous methods (TRADES, MART, and AWP) across the datasets**.
>     - FR and FR/WA can be plugged into these famous methods to further improve their robustness.
>
> 3. R4-2a (KL divergence): To compare the benefits of L1 and Kullback-Leibler (KL) divergence, we conduct experiments on CIFAR-10 (the popular dataset for AT) with a ResNet18 model. The KL divergence is applied to the real and imaginary parts of the complex numbers after FFT, respectively, and the results are summed (the same way as L1). Then we adjusted the FR coefficients to 0.001, 0.05, 0.1, 0.2, and 0.5, with the best performance being 0.05. The best robustness for FR with KL against PGD-20 and Autoattack are **53.48\%** and **48.92\%**, respectively. While FR with L1 achieves **55.39\%** and **50.02\%**, respectively.  KL focuses on the entire distribution, while L1 performs a one-to-one comparison. For the FFT-processed information, where the value of each pixel represents the magnitude of the corresponding frequency, a one-to-one comparison is more convincing than comparing frequency domain distributions. So **L1 loss is a better choice**. (In particular, the AT-CNNs focus primarily on low-frequency regions rather than on the overall frequency information.)
>
> 4. R4-2b (Failure cases): We refer to the successfully evaluated images as success cases and incorrectly classified images as failure cases. Taking the adversarially trained ResNet18 with FR on CIFAR-10 as an example, we visualize randomly selected images and the average Fourier spectra of all perturbations in Section 5 of the supplementary. The perturbations in failure cases are more concentrated in the low-frequency center than in success cases. This indicates that AT+FR gives the model a degree of resistance to perturbations at low- and medium-frequency regions. To further improve robustness, **it is essential to improve the resistance of the model to ultra-low-frequency perturbations**.
>
> [1] Jia, Xiaojun, et al. "LAS-AT: Adversarial Training with Learnable Attack Strategy." Proceedings of the IEEE/CVF Conference on Computer Vision and Pattern Recognition. 2022.
>
> [2] Wang, Yisen, et al. "Improving adversarial robustness requires revisiting misclassified examples." International Conference on Learning Representations. 2019.
>
> [3] Wu, Dongxian, Shu-Tao Xia, and Yisen Wang. "Adversarial weight perturbation helps robust generalization." Advances in Neural Information Processing Systems 33 (2020): 2958-2969.
>
> [4] Alayrac, Jean-Baptiste, et al. "Are labels required for improving adversarial robustness?." Advances in Neural Information Processing Systems 32 (2019).

---

### Official Review · Reviewer_j5uT · 2022-10-25

**Confidence:** 4
**Correctness:** 2
**Technical Novelty And Significance:** 2
**Empirical Novelty And Significance:** 2
**Recommendation:** 3

**Clarity, Quality, Novelty And Reproducibility:**

Clarity, quality, novelty and reproducibility are provided in Strength and Weakness Section.

**Strength And Weaknesses:**



Strength

- This paper describes the vulnerability of the neural network model against the white-box attack in terms of frequency distribution.
- The new regularization method called FR is proposed for improving the adversarial robustness.

Weakness / Question

- This paper has no theoretical result.
- In Table 1, I am confused how the values are computed. Can you explain the procedure with rigorous mathematical formuals if necessary? Also, why the Table 1 implies that AT focuses on low-frequency region? The robust model performs best for the highest bandwidth
- In Table 1, how is the last column evaluated by the learned model?
- The empirical results are not significant.
- The experimental setup for Tabel 3 is not well explained.
- WA can be applied to other adversarial algorithms. Thus, comparing the performance AT+FR/WA with other competitors is not fair. For fair comparison, WA should be applied to others.
- It seems that the result of AWP [2] in Table 3 is not reliable. The original paper reports the robust accuracy against AA is about 56. The 2% degration of performance in AutoAttack is largely significant. Can you explain about it?
- There is not enough explanation for model selection in the evalution setting.
- In eqn (2), what is the exact form of \mathcal{L}_{ce}? Does it depends on  f_1, f_2 or other f?
- In eqn (2), how can you find  in the regularization term? Is PGD in AT [1]? What is target model for finding \delta, f_1, f_2 or other model?
- In eqn (2), what \mathcal{F}(f_1(x)) means? It seems the outputs of FFT in logits. I'm confused what it means to apply FFT to logits. My understanding is that FFT is applied to data.
- The frequency regularization is not limited AT [1]. It can be extended to other algorithm such as TRADES. Can you present the results for other algorithms?
- I'm confused about what the L-model is. Is Train X of L-model LPF(X)?


[1] Madry, Aleksander and Makelov, Aleksandar and Schmidt, Ludwig and Tsipras, Dimitris and Vladu, Adrian, Towards Deep Learning Models Resistant to Adversarial Attacks, In ICLR, 2018.

[2]  Dongxian Wu, Shu-Tao Xia, and Yisen Wang, Adversarial Weight Perturbation Helps Robust Generalization, In NeurIPS, 2020.

**Summary Of The Paper:**

This paper observes that AT [1] focuses on the low-frequency area to achieving adversarial robustness. It empirically shows that white-box attack corrupts the high frequency domain. Authors propose Frequency Regularization (FR) which applies a regularization in the frequency domain with AT.

**Summary Of The Review:**

Authors empirically shows model trained by AT focuses on the low-frequency region for adversarial robustness and propose the Frequency Regularization. Attempts to explain adversarial learning in the frequency domain seem very interesting. But, it seems that the comparison with other competitors is not fair and empirical results are marginal.

---

> ### Author Response · Authors · 2022-11-18
> **Response to Reviewer j5uT**
>
> 9. R3-9 ($L_{CE}$): $L_{CE}$ is the traditional Cross-Entropy loss function. $L_{CE} = -\sum_{i=0}^{n-1} y_{i} log(p_{i})$, $n$ is the numbers of inputs, $y_{i}$ is the one-hot true label, $p_{i}$ is the predicted probability of $f_2$ network. $f_{2}$ is the final model that we evaluate the robust performance.
>
> 10. R3-10 (Regularization): Do you mean where we learned this regularization from? The FR is **originally proposed by us to improve the robustness**. Yes, the PGD is the same as [1]. The target model is $f_{2}$. As for the basic FR, $f_1, f_2$ are the same models in the AT.
>
> 11. R3-11 (FFT(output)): Your understanding is right. Typically, the FFT is applied to the images(data) to check the frequency distribution. Our motivation for applying FFT to logits is to make the model extract similar spectral features from the adversarial inputs as the natural ones, rather than focusing on the physical meaning. Through the backpropagation mechanism, this regularization will implicitly affect the feature maps extracted in the hidden layers.
>
> 12. R3-12 (TRADES+FR): Thanks for your advice. Since FR is **orthogonal** to TRADES, MART, and AWP, we apply FR to these methods to check their effectiveness. **The results shown in Section 3 of the supplementary prove that FR can further improve the robustness on top of these methods**.
>
> 13. R3-13 (L-model): L-model is a model **naturally trained with the inputs processed by an LPF** with a fixed bandwidth (half of the image size). $Input = LPF(x)$. $x$ denotes the natural image, $LPF$ denotes the low-pass filtering. Since only 1/4 of the low-frequency information is retained for the natural training, so we called it L-model.
>
> [1] Jia, Xiaojun, et al. "LAS-AT: Adversarial Training with Learnable Attack Strategy." Proceedings of the IEEE/CVF Conference on Computer Vision and Pattern Recognition. 2022.
>
> [2] Wang, Yisen, et al. "Improving adversarial robustness requires revisiting misclassified examples." International Conference on Learning Representations. 2019.
>
> [3] Wu, Dongxian, Shu-Tao Xia, and Yisen Wang. "Adversarial weight perturbation helps robust generalization." Advances in Neural Information Processing Systems 33 (2020): 2958-2969.
>
> [4] Alayrac, Jean-Baptiste, et al. "Are labels required for improving adversarial robustness?." Advances in Neural Information Processing Systems 32 (2019).

---

> ### Author Response · Authors · 2022-11-18
> **Response to Reviewer j5uT**
>
> We thank R3 for the detailed feedback.
> 1. R3-1 (theoretical result): Our work breaks the conventional belief that the perturbations are mostly high-frequency information. Furthermore, it demonstrates **for the first time** the self-adjusting property of the perturbations according to the model's frequency bias, which helps prove why it is challenging to defend white-box attacks.
>
> 2. -  R3-2a (Values in Table 1): The accuracy of Table 1 is calculated with an LPF processing the natural inputs: $\hat{y} = model (LPF_{d}(x))$, where $x$ is the natural input, and $\hat{y}$ is the model's prediction. The LPF is low-pass filtering with a bandwidth of $d$ listed in Table 1.
>     -  R3-2b (Focus on low frequency): Though the robust model performs best with the highest bandwidth, **the accuracy improvement is mainly brought by the utilization of low-frequency information**. Taking the CIFAR-10 as an example, the standard accuracies of the natural model with limited low-frequency information (d=16) and full information (d=32) are 46.71\% and 94.43\%, respectively. The improvement is **47.72\%**, which means the natural model utilizes **both** low- and high-frequency information for the predictions. For the AT-CNN, it achieves 78.63\% (much higher than 46.71\%) and 81.98\% standard accuracy concerning the limited low-frequency information (d=16) and full information (d=32), respectively. The improvement is only **3.35\%** (much smaller than 47.72\%), which implies that AT **mainly focuses on the low-frequency region**.
>
> 3. R3-3 (Last column in Table 1): The last column in Table 1 is the robustness against the PGD-20 attack to imply that simply focusing on the low-frequency information fails to bring the adversarial robustness (comparing the L- and robust models). First, we generate the adversarial perturbations $\delta = PGD(model, x, y)$, where $x, y$ are the natural input and the ground truth label. Then we feed the adversarial inputs into the model $\hat{y} = model (x + \delta)$ to compute the robust accuracy.
>
> 4. R3-4 (Results are not significant): It is widely perceived that raising adversarial robustness is a highly difficult task. LAS-AT[1] utilizes reinforcement learning. MART[2] revisited misclassified examples. None of these techniques applied to the AT give as high a boost as the FR in this paper. To improve robustness significantly, sophisticated techniques[3] and more data[4] are required. Our proposed method is **low in complexity and requires no additional data**. To this end, we proposed a **simple but effective** FR block. Taking the PGD-AT as the baseline, FR obtains higher robust accuracy without bells and whistles than some SOTA methods. In addition, there is a trade-off between the standard and the robust accuracies. The FR/WA method can **maintain the standard accuracy as the PGD-AT while improving the robust accuracy**, which is something nontrivial.
>
> 5. R3-5 (Experimental setup): Thanks for the note. The experimental setup for Table 3 follows the default settings in the original papers. Because of the word limit, we give the detailed settings in Section 2 of the supplementary.
>
> 6. R3-6 (Weight Averaging(WA)): We utilize the WA model to deal with the natural inputs shown in Figure 5 to form part of FR, which is quite **different from** the traditional WA (used for the final evaluation). In the equation: $L_{AT} = L_{CE} + \lambda \cdot \frac{1}{n} \sum_{i=1}^{n}Dis(\mathcal{F}(f_{1}(x_{i})), \mathcal{F}(f_{2}(x_{i}+\delta))$, WA model is denoted as $f_1$, viewed as a **teacher** to deal with the natural inputs. The model we evaluate the robust accuracy is denoted by $f_2$, viewed as a **student** dealing with adversarial inputs. We expect the teacher(WA) helps the student to extract useful information from the perturbed images, and **we evaluate the robustness of the student model, not the teacher(WA) model**.
>
>     Besides, we also apply WA to the standard AT on CIFAR-10 with a ResNet18 as the baseline. The robustness of **AT + WA** against PGD-20 and Autoattack are 51.82\% and 48.50\%, respectively. While the robustness of **AT + FR/WA** against PGD-20 and Autoattack are 55.12\% and 50.16\%, respectively. This indicates that **AT+FR/WA performs much better than AT+WA**.
>
> 7. R3-7 (AWP): Do you mean the TRADES-AWP method in Table 2 in the AWP paper? TRADES-AWP is a method combining TRADES and AWP. For a fair comparison, we compare the performance of famous methods **applied directly to the standard AT**, which is 54.04\% for AWP.
>
> 8. R3-8 (Model for evaluation): Thanks for your note. The training set was randomly divided into the training set and the validation set according to the ratio of 9:1. Following most papers, we selected the model with the **highest robustness against PGD-20 attacks** (common choice in papers[2][3]) on the validation set for further evaluation against other attacks on the test set.
>
> **Because of the word limit, follow-up questions are answered in the next comment.**

---

### Official Review · Reviewer_DY1G · 2022-10-31

**Confidence:** 3
**Correctness:** 3
**Technical Novelty And Significance:** 2
**Empirical Novelty And Significance:** 2
**Recommendation:** 5

**Clarity, Quality, Novelty And Reproducibility:**

The paper is clearly written, except that there is too much discussion on a well accepted fact.
The method seems novel to me. For reproducibility, I cannot judge but it seems no problem for the reported result.

**Strength And Weaknesses:**

strength:
+ as far as I know, the proposed method that aligns frequency domain response is novel.
+ in numerical experiments, the frequency regularization shows advantages over other methods.

weakness:
- it is well known and can be understood that adversarial perturbations are mainly on high-frequency part. Thus, it is not necessary to provide toy examples for this point, i.e., Section 3.2 could be largely reduce.
- some researchers think that it is better to use more flexible model for AT, since essentially we have two distributions to fit. In this point of view, it is better to enhance the flexibility, not to restrict it by regularization. Maybe for small model, regularization helps but for large model it becomes useless. I would like to know the opinion of the authors and if there are additional experiments on large model, it will be great.
- AT is actually a trade-off between natural and adversarial samples. I notice that the proposed method generally has lower clean accuracy, from which it follows a question: how about the robustness when the clean accuracy is similar. Then the comparison could be more interesting.

**Summary Of The Paper:**

Based on the fact that adversarial perturbations are mainly on high-frequency part, this paper propose a regularization term to align the frequency response of natural and adversarial samples.

**Summary Of The Review:**

The motivation of introducing frequency regularization is not surprising. But the performance is indeed good. I hope I could receive additional explanation or result for the questions I raised above.

---

> ### Author Response · Authors · 2022-11-18
> **Response to Reviewer DY1G**
>
> We thank R2 for the valuable and thoughtful feedback.
>
> 1. R2-1 (Perturbations are high-frequency): We apologize for any misunderstanding due to our presentation. Indeed, our study revamps the perception of perturbations as high-frequency information, and demonstrates an adversary will self-adjust according to the model's frequency bias. Previous works [1,2] have discovered that, for **naturally trained** models, the adversarial perturbations mainly focus on the high-frequency regions. These works, however, **did not** investigate the adversarial perturbations of **adversarially trained** models. To fill this void, our study goes further to explore the frequency distribution across models and datasets, discovering that the perturbations are **not an invariant high-frequency phenomenon**, but are data- and model-dependent. This is one of our contributions, and we also provide the visualization of randomly selected images across the datasets in Section 1 of the supplementary to further prove it.
>
>     In fact, our paper discovers that the frequency distribution of adversarial perturbations is not invariant. As in Figure 2 in the main text, we give the frequency distribution of perturbations across different models and datasets, proving adversarial perturbations are **not** simply targeting high-frequency regions. To our knowledge, our work is the **first-ever** to show that perturbations can adapt their frequency distribution to the model's frequency bias. For example, L-models and adversarially-trained (robust) models focus more on low-frequency information for predictions, so the perturbations' frequency distribution is mainly concentrated on the **low-frequency region**. Based on this observation, we dig deeper into the frequency distribution of perturbations to explain why white-box attacks are difficult to defend and propose Frequency Regularization to improve adversarial robustness.
>
> 2. R2-2 (Two distributions? large model): Thanks for your nice advice. We check whether the natural and adversarial examples are two distributions and conduct experiments to prove the effectiveness of the proposed methods on large models.
>
>     - We check if the clean and adversarial examples are two different distributions by computing the mean and standard deviation. Using the CIFAR-10 test set as an example, we calculated the mean and standard deviation of three channels of the clean dataset as: [0.4942 0.4851, 0.4504] and [0.2466, 0.2428, 0.2615], respectively. The mean and standard deviation of the adversarial examples generated according to the natural model are [0.4941, 0.4850, 0.4501] and [0.2464, 0.2425 0.2609], and the means and variances generated from the robust model are [0.4940, 0.4853, 0.4502] and [0.2454, 0.2418, 0.2602]. The maximum difference of the means and standard deviation between the clean and adversarial examples are only **0.0002** and **0.0013**. So we can see the distributions of clean and adversarial samples are in fact **quite close**.
>
>     - Experimental results on large models are shown in Section 4 of the supplementary. The robustness against popular PGD and AutoAttack proves that the FR and FR/WA blocks are also **suitable for large models** (e.g., ResNet152, Wide ResNet-34-15).
>
> 3. R2-3 (similar standard accuracy): AT+FR can further improve robustness on top of AT with a small decrease in standard accuracy. Taking ResNet18 trained at CIFAR-10 as an example, the robustness is the accuracy of responding to PGD-20 attack. The most robust **AT** model achieves **81.98\%** clean accuracy and **51.69\%** robust accuracy. AT + FR is not able to achieve a similar clean accuracy, the highest clean accuracy achieved by it is 81.02\%, and its robustness is 54.27\%. To overcome this problem, we optimize the FR with a WA block to deal with the natural inputs. **AT + FR/WA** achieves **81.74\%** clean accuracy and **55.12\%** robust accuracy. Results in Tables 2 and 3 in the manuscript show that **FR/WA can maintain the clean accuracy as the PGD-AT while improving the robustness, which is something nontrivial.**
>
>     Besides, all methods based on AT cannot achieve a similar clean accuracy as natural training. Ref.[1] claimed that naturally trained CNNs need to pick up high-frequency information to achieve higher training accuracy. Therefore, we attribute this degradation to the fact that AT-CNNs focus mainly on low-frequency information.
>
>
> [1] Wang, Haohan, et al. "High-frequency component helps explain the generalization of convolutional neural networks." Proceedings of the IEEE/CVF Conference on Computer Vision and Pattern Recognition. 2020.
>
> [2] Wang, Haohan, et al. "High-frequency component helps explain the generalization of convolutional neural networks." Proceedings of the IEEE/CVF Conference on Computer Vision and Pattern Recognition. 2020.

---

> > ### Comment · Reviewer_DY1G · 2022-11-21
> > **thanks for the reply**
> >
> > Thanks for the reply.
> >
> > For the discussions on ``two distributions'', simply calculating the distance of features extracted by DNNs is not convincing, since there are already nonlinear mapping related to labels. If the two distributions are very close, or the same, then AT will not hurt the clean accuracy, which however is not true in practice.
> >
> > For the question ``similar standard accuracy'',  from Table 2 and Table 3, the clean accuracy of AT+FR (/WA) is quite low and then it is not easy to show the advantage, since it could be explained as a trade off.  I think the best way to show their advantage is to adjust their clean accuracy a bit higher than others and then compare the robustness.

---

> > > ### Author Response · Authors · 2022-11-22
> > > **Response to Reviewer DY1G**
> > >
> > > Thanks for your reply and useful suggestions. We try to respond as follows.
> > > 1. R2-1 (Two distributions)：Regarding "simply calculating the distance of features extracted by DNNs". We are sorry that we did not make it clear enough. Our statistics for the mean and standard deviation in the rebuttal are computed on the natural and adversarial **input images**, **not on the deep features** extracted by the CNNs. What we wanted to show is that the natural and adversarial inputs can be both visually and statistically indistinguishable. And that as is well known, a slight difference in input images can lead to a large output discrepancy. (e.g. in Su et.al. ``One pixel attack for fooling deep neural networks'', modifying just one pixel of the image can significantly reduce the accuracy).
> > >
> > >     On a side but related note, we remark that the low standard accuracy of AT is due to the fact that AT guides the model to **rely mainly on low-frequency information for prediction**. Without the utilization of high-frequency information, the standard accuracy of AT model cannot reach as high as the natural model. So the degradation in the standard accuracy is **not caused by the difference of the distributions, but the model’s attention**. Moreover, as may be seen from Table 1 in the manuscript, the natural accuracy of the L-model (trained with low-frequency information) is also lower than that of the natural model under the same natural inputs from the test set.
> > >
> > > 2. R2-2(Similar standard accuracy): Thanks for your useful comment, and we understand your point. We spent a day running such experiments with ResNet18 on CIFAR-10, and the Table below shows the results. ($\epsilon$ represents the $l_{\infty}$ norm limitation to the perturbations. We only modified the training attack hyperparameter, and the test one was not modified.)
> > >
> > >     | Method | $\epsilon$ | Clean | PGD-20 | PGD-50 | C&W | AA |
> > >     | --- | --- | --- | --- | --- | --- | --- |
> > >     | Natural training | 8/255 | 94.43 | 0.0 | 0.0 | 0.0 | 0.0 |
> > >     |PGD-AT | 8/255 | 81.98 | 51.69 | 51.46 | 50.44 | 48.19 |
> > >     |TRADES | 8/255 | 81.83 | 53.41 | 53.23 | 50.92 | 49.84 |
> > >     |MART | 8/255 | 81.01 | 54.58 | 54.47 | 50.01 | 48.10 |
> > >     |AWP | 8/255 | 81.06 | 55.36 | 55.27 | 51.98 | 50.37 |
> > >     | AT + FR/WA | 8/255 | 81.74 | 55.12 | 54.88 | 52.21 | 50.16 |
> > >     | AT + FR/WA | 4/255 | 87.67 | 44.23 | 43.76 | 42.87 | 40.17 |
> > >     | AT + FR/WA | 2/255 | 90.37 | 35.43 | 35.02 | 33.98 | 31.42 |
> > >
> > >     Note that by weakening the attack strengths (bottom two rows) we can raise the clean accuracy of the proposed AT+FR/WA scheme to over 90\%. As expected, this comes at the expense of reduced robustness, yet we can still observe uniform and decent robustness across all attacks, an evidence proof that AT+FR/WA is effective and generalizable. Again, under the same attack strength (3rd row from bottom), AT + FR/WA has only a **0.24% decrease in clean accuracy and a 3.43% increase in robustness** against PGD-20 **relative to PGD-AT**. Although trade-off exists, the gains outweigh the losses for the goal of improving robustness. More experimental results of applying FR/WA to other defenses to maintain similar clean accuracy and further improve robustness can be viewed in section 3 of the supplementary.
> > >
> > >
> > >     Last but not least, in the context of adversarial learning study, the **baseline** for the natural accuracy is generally **the accuracy of the AT models**. TRADES, MART, and AWP are all dedicated to improving robustness, and their natural accuracies are essentially similar to AT. To this end, our proposed AT+FR/WA can maintain the standard accuracy as regular AT and improves **more robustness** than these defenses.

---

> > > ### Author Response · Authors · 2022-11-23
> > > **Response to Reviewer DY1G**
> > >
> > > As an add-on to our previous reply, also as a further response to the reviewer's query, let us give another perspective that may inspire further.
> > >
> > >   R2-2 (Similar standard accuracy): Specifically, instead of ablating through strong and weak attacks, we fix a strong attack strength and observe the network robustness across training epochs. To begin with, the models in Tables 2 and 3 in the paper are selected with the **highest robust accuracy** as we aimed for the most robust models. But if we want to optimize for a higher clean accuracy, we can pick several checkpoints **with a higher clean accuracy to check its robust performance**. Taking the large Wide ResNet-34-10 on CIFAR-10 as an example, the table below shows that the proposed FR/WA can achieve a higher standard and robust accuracies than other defenses (2nd row of AT+FR/WA better than TRADES and AWP, 1st row of AT+FR/WA better than MART, bottom row of AT+FR/WA has a much higher clean accuracy than all others and still a higher robustness than PGD-AT). This further proves the effectiveness of the proposed FR/WA over other SOTA defenses. To summarize, AT can raise robust accuracy at the expense of the standard accuracy, viz. a trade-off. But some checkpoints trained with FR/WA can  improve **both the standard and robust accuracies** relative to AT, which is nontrivial. (Bold numbers in the table indicate the best.)
> > >
> > > | Method | $\epsilon$ | Clean | PGD-20 | PGD-50 | C&W | AA |
> > > | --- | --- | --- | --- | --- | --- | --- |
> > > | PGD-AT | 8/255 | 84.62 | 55.01 | 54.88 | 53.32 | 51.42 |
> > > | TRADES | 8/255 | 84.92 | 56.33 | 56.13 | 54.20 | 53.08 |
> > > | MART | 8/255 | 84.17 | 58.56 | 58.06 | 54.58 | 51.10 |
> > > | AWP | 8/255 | 85.57 | 58.14 | 57.92 | 55.96 | 54.04 |
> > > |  |  |  |  |  |  |  |
> > > | AT + FR/WA | 8/255 | 84.57 | **59.27** | **58.90** | **56.92** | **54.80** |
> > > | AT + FR/WA | 8/255 | 86.53 | 58.39 | 58.12 | 56.14 | 54.07 |
> > > | AT + FR/WA | 8/255 | 87.78 | 57.47 | 57.20 | 55.25 | 53.28 |
> > > | AT + FR/WA | 8/255 | **89.12** | 55.34 | 55.02 | 53.42 | 51.47 |

---

### Official Review · Reviewer_q8SJ · 2022-11-05

**Confidence:** 4
**Correctness:** 2
**Technical Novelty And Significance:** 2
**Empirical Novelty And Significance:** 2
**Recommendation:** 5

**Clarity, Quality, Novelty And Reproducibility:**

The paper is clear.
The results are reproducible.
Paper lacks the required novelty for ICLR publication.


**Strength And Weaknesses:**

Strength
1-- The paper is well-written.
2-- The paper studies an important problem.
3-- The proposed method is explained clearly.

Weaknesses:
1--It is well known in the literature that adversarial perturbations are targeting high-frequency.
2-- The proposed training does not provide major new findings.


**Summary Of The Paper:**

The authors investigate AT from a spectral perspective and show that AT induces the deep model to focus more on the low-frequency region.
They find that the spectrum of a white-box attack is distributed in regions the model focuses on. They propose a frequency regularization such that the spectral output inferred by an attacked input stays close to its natural input.

**Summary Of The Review:**

Please see the comments above.

---

> ### Author Response · Authors · 2022-11-18
> **Response to Reviewer q8SJ**
>
> We thank R1 for the response.
>
> 1. R1-1 (Perturbations are high-frequency): We apologize for any misunderstanding due to our presentation. Indeed, our study revamps the perception of perturbations as high-frequency information, and demonstrates an adversary will self-adjust according to the model's frequency bias. Previous works [1,2] have discovered that, for **naturally trained** models, the adversarial perturbations mainly focus on the high-frequency components. These works, however, **did not** investigate the adversarial perturbations of **adversarially trained** models. To fill this void, our study goes further to explore the frequency distribution across models and datasets, discovering that the perturbations are **not an invariant high-frequency phenomenon**, but are data- and model-dependent. This is one of our contributions, and we also provide the visualization of randomly selected images across the datasets in Section 1 of the supplementary to further prove it.
>
>     In fact, our paper discovers that the frequency distribution of adversarial perturbations is not invariant. As in Figure 2 in the main text, we give the frequency distribution of perturbations across different models and datasets, proving adversarial perturbations are **not** simply targeting high-frequency regions. To our knowledge, our work is the **first-ever** to show that perturbations can adapt their frequency distribution to the model's frequency bias. For example, L-models and adversarially-trained (robust) models focus more on low-frequency information for predictions, so the perturbations' frequency distribution is mainly concentrated on the **low-frequency region**. Based on this observation, we dig deeper into the frequency distribution of perturbations to explain why white-box attacks are difficult to defend, and propose Frequency Regularization to improve adversarial robustness.
>
> 2.  R1-2 (No new findings):  Thanks for your comments. We would like to further explain and stress the novelty and contributions of our work. It is well known that improving adversarial robustness is a challenging problem. For example, under the attack by PGD-20, even the famous TRADES framework can only improve the robustness by less than 2\% compared to the standard AT.
>
>     Firstly, we observe **for the first time** that the spectrum of a white-box attack is primarily distributed in regions the model focuses on, and the perturbation attacks the spectral bands where the model is vulnerable. Next, we propose two **simple yet effective methods**, namely, Frequency Regularization (FR) and FR with Weight Averaging (FR/WA) to boost robust accuracy.
>
>     - FR: For standard adversarial training, FR is a plug-and-play module. As in Table 2 in the manuscript, it can improve the adversarial robustness against several attacks across datasets. For the large model Wide ResNet-34-10, it improves more robustness than other famous methods (cf. Table 3). More experiments are added in Section 3 of the supplementary to prove that plugging FR into popular defenses can further enhance robustness.
>
>     - FR/WA: Weight Averaging (WA) is commonly used to average the weight values over time (epochs) along the training trajectory **for the final evaluation**. However, it is employed in our method to deal with the natural inputs shown in Figure 5 in the manuscript, which is significantly different from the conventional WA.  Specifically, for the FR equation: $L_{AT} = L_{CE} + \lambda \cdot \frac{1}{n} \sum_{i=1}^{n}Dis(\mathcal{F}(f_{1}(x_{i})), \mathcal{F}(f_{2}(x_{i}+\delta))$, WA model is $f_1$, viewed as a **teacher** to deal with the natural inputs. The model that we evaluate the robust accuracy is $f_2$, viewed as a **student** to deal with the adversarial ones. We hope the teacher (WA) helps the student to extract useful information from the perturbed images, and we evaluate the robustness of the student model **not** on the teacher (WA) model.
>
>
>
> [1] Wang, Zifan, et al. "Towards frequency-based explanation for robust cnn." arXiv preprint arXiv:2005.03141 (2020).
>
> [2] Wang, Haohan, et al. "High-frequency component helps explain the generalization of convolutional neural networks." Proceedings of the IEEE/CVF Conference on Computer Vision and Pattern Recognition. 2020.

---

### Author Response · Authors · 2022-11-28
**Additional comments, if any, are most welcome and appreciated**

We would like to thank you again for your thoughtful reviews and valuable feedback.

We would appreciate it if you could let us know if our responses have addressed your concerns and whether you still have any other questions about our rebuttal.

We would be happy to do any follow-up discussion or address any additional comments.

---

### Decision · Program_Chairs · 2023-01-20

**Decision:**

Reject

**Justification For Why Not Higher Score:**

Based on reviewers' comments, the paper is considered to need more work to clarify the contributions and differences between FR and existing works.
Most importantly, the robustness evaluation is insufficient as the adaptive attack and the recent well-known auto attack (AA) were not considered in this paper.

**Justification For Why Not Lower Score:**

N/A

**Metareview: Summary, Strengths And Weaknesses:**

Based on the fact that adversarial training (AT) has proven to be an effective defense approach and the onservation that the properties of AT for robustness improvement remain an open issue, the authors investigate AT from a spectral perspective, providing new insights into the design of effective defenses. The proposed method, called Frequency Regularization (FR) is proposed for improving the adversarial robustness.
Overall, the reviewers raised the common concern in the first round of reviews regarding the well-known finding that the adversarial perturbations are targeting high-frequency.
Based on reviewers' comments, the paper is considered to need more work to clarify the contributions and differences between FR and existing works.
Most importantly, the robustness evaluation is insufficient as the adaptive attack and the recent well-known auto attack (AA) were not considered in this paper.
This paper is recommended to be rejecgted at its current status.